# On Some Properties of the Glacial Isostatic Adjustment Fingerprints

**Giorgio Spada** [1,*,†] **and Daniele Melini** [2,†]

1   Dipartimento di Scienze Pure e Applicate (DiSPeA), Sezione di Fisica, Università degli Studi di Urbino "Carlo Bo", I-61029 Urbino, Italy

2   Istituto Nazionale di Geofisica e Vulcanologia (INGV), Via di Vigna Murata 605, I-00143 Rome, Italy

*   Correspondence: giorgio.spada@gmail.com

†   These authors contributed equally to this work.

**Abstract:** Along with density and mass variations of the oceans driven by global warming, Glacial Isostatic Adjustment (GIA) in response to the last deglaciation still contributes significantly to present-day sea-level change. Indeed, in order to reveal the impacts of climate change, long term observations at tide gauges and recent absolute altimetry data need to be decontaminated from the effects of GIA. This is now accomplished by means of global models constrained by the observed evolution of the paleo-shorelines since the Last Glacial Maximum, which account for the complex interactions between the solid Earth, the cryosphere and the oceans. In the recent literature, past and present-day effects of GIA have been often expressed in terms of *fingerprints* describing the spatial variations of several geodetic quantities like crustal deformation, the harmonic components of the Earth's gravity field, relative and absolute sea level. However, since it is driven by the delayed readjustment occurring within the viscous mantle, GIA shall taint the pattern of sea-level variability also during the forthcoming centuries. The shapes of the GIA fingerprints reflect inextricable deformational, gravitational, and rotational interactions occurring within the Earth system. Using up-to-date numerical modeling tools, our purpose is to revisit and to explore some of the physical and geometrical features of the fingerprints, their symmetries and intercorrelations, also illustrating how they stem from the fundamental equation that governs GIA, i.e., the Sea Level Equation.

**Keywords:** Glacial Isostatic Adjustment; sea level change; fingerprints of past ice melting

## 1. Introduction

Since the Glacial Isostatic Adjustment (GIA) caused by the melting of past ice sheets is still contributing to present-day regional and global sea-level variations, understanding its spatio-temporal variability is important to interpret the effects of present climate change. However, because GIA is also affecting the shape of the Earth and its gravity field, it has a clearly established role in the interpretation of a wide range of geophysical data, obtained by means of ground-based or space-borne geodetic observations. The relevance of GIA modeling in various fields of the Geosciences and its increasing role in the framework of global change constitutes the motivation of this study, mainly aimed at reviewing systematically the imprints of GIA and to describe their geometry in connection with the Sea Level Equation (SLE).

To introduce GIA, it is convenient to define a *reference state* in which the solid Earth, the ice sheets and the oceans are in an equilibrium configuration, sketched in Figure 1a, and to compare it to a subsequent perturbed state. This approach was originally proposed by Farrell and Clark [1], hereafter referred to as FC76, in their seminal work where the SLE was introduced first. The reference

configuration can be chosen arbitrarily, but for our discussion it is convenient to refer to the Last Glacial Maximum (LGM, $\sim$21,000 years ago). The load acting on the Earth's surface in the reference state (i.e., the mass per unit area) is $L_0(\omega)$ and $I_0(\omega)$ is the ice thickness, with $\omega = (\theta, \lambda)$ where $\theta$ is colatitude and $\lambda$ is longitude. The SLE has can be employed to predict how sea level shall change at an arbitrary location $\omega$, when the configuration of the system portrayed in Figure 1a evolves in a *new state* shown in Figure 1b at time $t \geq t_0$, in which the surface load and the ice thickness are $L(\omega, t)$ and $I(\omega, t)$, respectively. Despite the global variations observed in the new state, (i) the mass of the system (ice+oceans+solid Earth) must be conserved, and (ii) the new sea surface must remain an equipotential; ultimately, these are the two fundamental principles that the SLE makes manifest.

The interactions responsible for the changes observed in the new state are qualitatively sketched in the diagram of Figure 2, freely modified from Clark et al. [2]. Since the interactions are operating simultaneously and at all spatial scales, their contributions cannot be easily disentangled, which makes the interpretation of the GIA effects on sea level and on various geodetic quantities particularly challenging. In the top part, the figure is showing the three fundamental elements of the SLE, i.e., the ice sheets, the solid Earth, and the oceans [3]. As indicated by the arrows, these elements are interacting by two mechanisms: (i) surface loading and (ii) mutual gravitational attraction.

The waxing and waning ice sheets, by changing the local ice masses, exert a load at the surface of the solid Earth (*ice loading*, related to glacio-isostasy), but the mass variation of the oceans is also loading the Earth, acting on the seafloor (*water loading*, associated to hydro-isostasy). These two non-uniform loads are tightly interconnected, since the mass conservation of the system (water + ice) imposes that, on average, the load variation vanishes across the Earth's surface. Due to the mantle imperfect elasticity, the  past loads also induce delayed and still persistent effects that are manifest as a global state of isostatic disequilibrium. Furthermore, the equipotential surfaces of the Earth's gravity field are twisted by the mass redistributed over the solid Earth and in the oceans, causing variations of the geoid. The three elements that enter into the SLE are all affected by gravitational attraction. In particular, the sea surface is warped by the attraction of the continental ice sheets, but at the same time the geoid variations caused by the solid Earth deformation modify the shape of the oceans. The bottom part of Figure 2 considers further interactions driven by the Earth's irregular rotation. Inertia perturbations, associated to long wavelength deformations and sea-level variations of harmonic degree $l = 2$, drive excursions of the rotation axis in order to conserve the Earth's angular momentum [4]. The consequent variation of the centrifugal potential alters, in turn, both the solid Earth and the sea surface, a process that is known as *rotational feedback* on sea level, see Peltier [5].

The *inextricably related interactions* first acknowledged by Clark et al. [2] and illustrated in Figure 2 are responsible for the regional imprints of GIA. As first noted by Woodward [6] and later discussed by Daly [7], Walcott [8] and Farrell and Clark [1], the sea-level variations associated with glacial isostasy depart significantly from the spatially uniform pattern that we would observe for a rigid, non-gravitating and non-rotating Earth (i.e., ignoring the interactions). Often, in the geological literature the spatially uniform sea-level change is referred to as *eustatic*, a word attributed to Suess [9]; eustatic variations only depend on the history of the past grounded ice volume [10]. Presently, the term *barystatic* is preferred [11]. The interactions are responsible for a global pattern of relative sea level (RSL) variations during the melting of the late-Pleistocene ice sheets, which Clark et al. [2] have characterized by defining six *RSL zones*, labelled from $I$ to $VI$ (see their Figure 5); within each zone, the sea-level signatures are similar to one another. The RSL zones encompass the glaciated areas (zone $I$), the region of the collapsing fore-bulge ($II$), the time-dependent emergence ($III$) and the oceanic submergence zone ($IV$), the  oceanic emergence region ($V$), and the continental shorelines ($VI$). Subsequently, Mitrovica and Milne [12] have studied the nature of the RSL zones in connection with the various terms of the SLE, describing the physical mechanisms responsible for their establishment and unveiling the processes of *continental levering* and *ocean siphoning*. Following the above studies, the spatial variability in sea level associated with GIA has been widely investigated with the aim of reconstructing the history of deglaciation since the LGM (see e.g., [13–15]). On a more limited spatial

scale, the concept of RSL zone has also been useful to interpret the Holocene sea-level variations across the Mediterranean Sea [16,17].

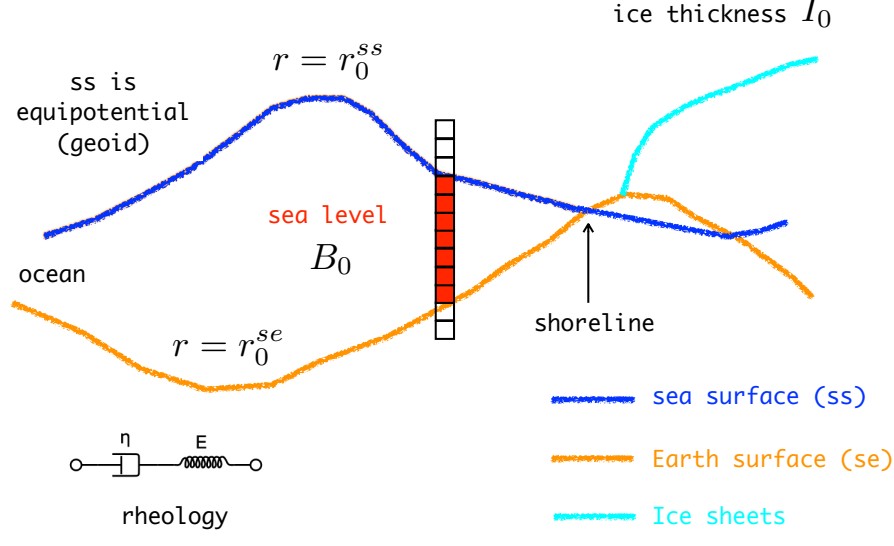

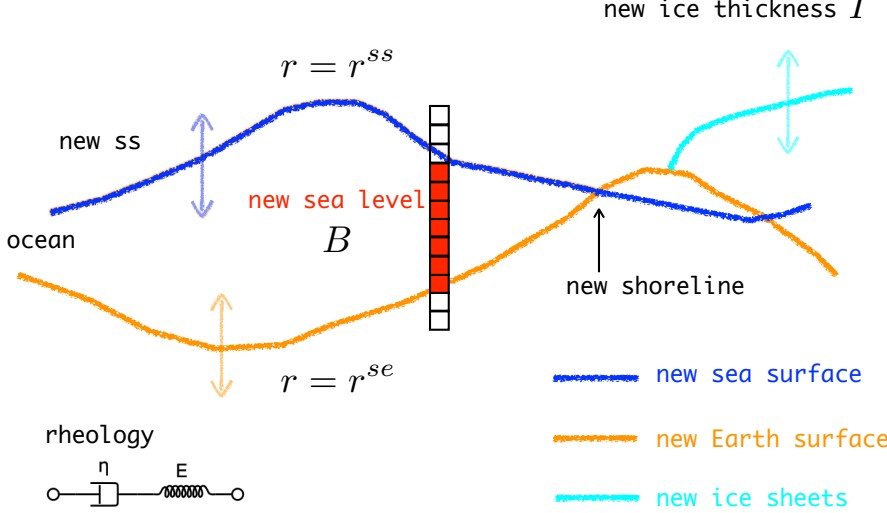

**Figure 1.** Sketches of the reference state for time $t = t_0$ (**a**) and of the general configuration for $t \geq t_0$ (**b**) showing the three Earth's portions that are interacting in the SLE: the solid Earth, the oceans and the ice sheets. With $r$ we denote the radius of a given point relative to the center of mass of the whole Earth system. Changes in sea level relative to the solid Earth are observed by the red stick meter. The sea surface is equipotential in (**a**) but also in (**b**), after that the ice sheets have shrunk and the mass of the oceans has consequently varied to compensate exactly the ice mass loss. The vertical arrows in (**b**) indicate that the sea surface and the solid Earth have moved relative to the origin of the reference frame. The spring-dashpot system is an exemplification of Earth's rheology.

The study of paleo-shorelines has allowed to define the broad features of the pattern of RSL zones since the LGM (see, e.g., Lambeck and Chappell [18]). However, the present-day trends of sea level detected at tide gauges or by satellite altimetry should be certainly also affected by contemporary variations in the state of the cryosphere driven by global warming. In this context, the question has not been addressed until the work of Plag and Jüettner [19], who have first coined the term of

*fingerprint* (function). Quoting their same words, ...*The elastic response of the Earth to present-day changes in the cryosphere can be expected to produce a similar fingerprint, which should be present in the tide gauge data. Based on these fingerprints, tide gauge trends, in principle, can be inverted for ice load changes* [19]. However, after having analyzed the relative sea-level trend for some long tide gauge time series, Douglas [20] concluded that *unambiguous evidence for fingerprints of glacial melting was not found, most likely due to the presence of other signals present in sea-level records that cannot easily be distinguished.* Recently, Spada and Galassi [21] have quantitatively compared the harmonic power spectrum of contemporary sea-level change to that of GIA, including the contribution due to the disintegration of the past ice sheets and that associated to present deglaciation. They have shown that the *power* of GIA from past ice melting is comparatively modest at all harmonic degrees, with the possible exception of harmonic degree $l = 2$, and it cannot emerge from the steric component that dominates current sea-level rise [22]. Notwithstanding the difficulty of visualization, the concept of sea-level fingerprint has undoubtedly gained an important role in the interpretation of the trends of contemporary [23–27] and future sea-level rise [28–30].

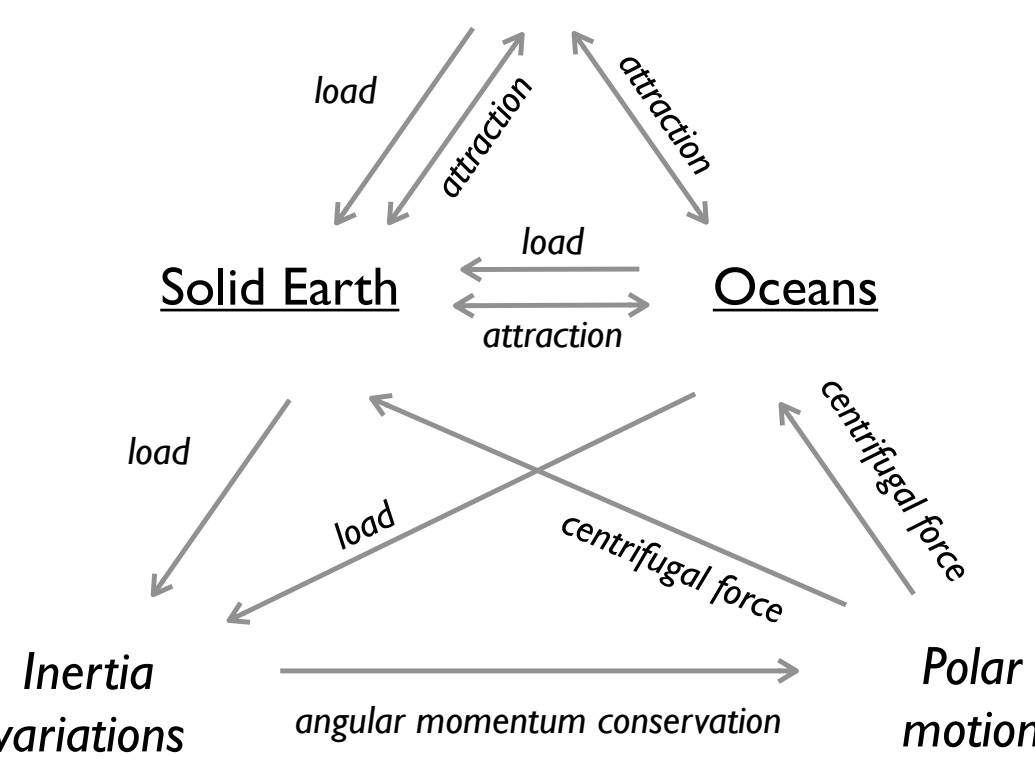

**Figure 2.** The top part of the triangle shows the elements that are perpetually interacting in the SLE (the solid Earth, the ice sheets and the oceans) through surface loading and mutual gravitational attraction. The bottom part qualitatively shows how Earth rotational effects are coming into play. The figure is inspired to that originally published by Clark et al. [2].

In this work, we aim at exploring and reviewing the properties and the symmetries of the GIA fingerprints presently associated with the melting of past ice sheets, as well as the intercorrelations among them. Much of what we present in this paper can be also applied to the fingerprints of present ice melting, which obey the SLE as well; these have been discussed in various places, see e.g., [31] and references therein. Although we are aware that uncertainties on the Earth's viscosity profile and the chronology of deglaciation affect significantly the pattern and the amplitude of the GIA fingerprints [32], here for simplicity we shall only consider a specific GIA model, leaving an error

analysis to future work, along the lines of Melini and Spada [32]. The paper is organized as follows. In Section 2 we review the theory behind the SLE. In Section 3 we briefly present the GIA model used and the numerical approach adopted. In Section 4 we illustrate some of the properties of the present-day GIA fingerprints associated with the melting of the past ice sheets, which in Section 5 are exploited to interpret the global uplift pattern of continents currently detected by GPS data. Our conclusions are drawn in Section 6.

## 2. Theory

Here we briefly introduce the essentials of the SLE theory, necessary to illustrate the geometry of the GIA fingerprints in Section 4 below. The reader is referred to Spada and Melini [33] (hereinafter SM19) and to its supplement for a more detailed and self-contained presentation. The paper SM19 and its supplement are presently (August 2019) submitted to Geoscientific Model Development (GMD), the interactive open-access journal of the European Geosciences Union at https://www.geosci-mod el-dev-discuss.net/gmd-2019-183/. The open-source program SELEN[4] (SELEN version 4.0) can be obtained from https://zenodo.org/record/3377404. We note that the SLE theory does not account for tectonic deformations nor for variations in the temperature or salinity of the ocean water, which we do not consider in our analysis.

In the reference state considered in Figure 1a, *sea level* is defined by the difference

$$B_0(\omega) = r_0^{ss} - r_0^{se}, \tag{1}$$

where $\omega = (\theta, \lambda)$ are the coordinates of a given point on the Earth's surface, while $r_0^{ss}(\omega)$ and $r_0^{se}(\omega)$ are the radii of the (equipotential) sea surface and of the solid Earth in a geocentric reference frame with origin in the whole-Earth center of mass, respectively. As shown in Figure 1a, $B_0$ would be directly measured by a stick meter, i.e., a *tide gauge*, placed at $\omega$. Assuming that the horizontal displacement of the stick-meter has been negligible in comparison to vertical displacement, in the new state sea level is

$$B(\omega, t) = r^{ss} - r^{se}, \tag{2}$$

where $r^{ss}(\omega, t)$ and $r^{se}(\omega, t)$ denote the new radius of the equipotential sea surface and of the solid surface of the Earth, respectively. Note that *topography* is related to sea level through

$$T(\omega, t) = -B. \tag{3}$$

Combining (2) with (1), *relative sea-level change*

$$\mathcal{S}(\omega, t) = B - B_0 \tag{4}$$

can be also expressed as

$$\mathcal{S}(\omega, t) = \mathcal{N} - \mathcal{U}, \tag{5}$$

where

$$\mathcal{N}(\omega, t) = r^{ss} - r_0^{ss} \tag{6}$$

is the *sea surface* variation, or *absolute sea-level change*, and

$$\mathcal{U}(\omega, t) = r^{se} - r_0^{se} \tag{7}$$

is the *vertical displacement* of the Earth's surface. Equation (5) represents the most basic form of the SLE. We note that, being defined as a double difference, relative sea-level change $\mathcal{S}(\omega, t)$ is not dependent upon the choice of the origin of the reference frame, i.e., it is an absolute quantity. Quantities $\mathcal{N}(\omega, t)$ and $\mathcal{U}(\omega, t)$, however, depend on the choice of the origin.

The sea surface variation $\mathcal{N}(\omega,t)$ is tightly associated to the variation of the geoid height. However, as remarked by FC76, $\mathcal{N}(\omega,t)$ is not *the* variation of the geoid height: i.e., *on a rigid Earth model, there is no distinction between changes in geoid radius and changes in sea level, but it is important to realize the difference between these quantities for deformable Earth models* [1]. A further problem arises from the fact that, in the new state, the volume of the oceans is varied to compensate the mass lost or gained by the continental ice sheets. Indeed, as pointed out by Tamisiea [34], some confusion arose recently about the definition of $\mathcal{N}(\omega,t)$, which sometimes is still used as a synonymous of geoid height variation; the confusion is attributed to often inconsistent terminology between various disciplines. FC76 have shown that the sea surface height variation is

$$\mathcal{N}(\omega,t) = \mathcal{G} + c, \tag{8}$$

where

$$\mathcal{G}(\omega,t) = \frac{\Phi}{g}, \tag{9}$$

is the *variation of the geoid* radius relative to the reference state, $\Phi(\omega,t)$ is the variation of the total gravity potential of the Earth system, taking both surface loading and rotational contributions into account, $g$ is the reference surface gravity acceleration and $c$ is a yet undetermined spatially invariant term, notorious within the GIA community as the *FC76 c-constant*. In the following, Equation (8) shall be referred to as *FC76 formula*. Thus, using Equation (8) in (5), the SLE reads

$$\mathcal{S}(\omega,t) = \mathcal{R} + c, \tag{10}$$

where we have defined the *sea-level response function* as the difference

$$\mathcal{R}(\omega,t) = \mathcal{G} - \mathcal{U}. \tag{11}$$

It is now convenient to average both sides of Equation (10) over the oceans, where the ocean-average of any function $F(\omega,t)$ is defined, at time $t$, as

$$< F(\omega,t) >^o (t) \equiv \frac{1}{A^o} \int_o F(\omega,t)\, dA, \tag{12}$$

where $\int_o$ denotes the integral over the time-dependent surface of the oceans, $A^o$ is their area at time $t$, $dA = a^2 \sin\theta d\theta d\lambda$ is the element of area over the surface of the sphere, and $a$ the average Earth's radius. We recall that the surface of the oceans is the region where $O = 1$, where $O$ is the *ocean function* (OF) defined as

$$O(\omega,t) = \begin{cases} 1, & \text{if } T + \dfrac{\rho^i}{\rho^w}I < 0 \\[2mm] 0, & \text{if } T + \dfrac{\rho^i}{\rho^w}I \geq 0, \end{cases} \tag{13}$$

where $\rho^i$ and $\rho^w$ are the densities of ice and water, respectively. For $O = 1$, the ocean is ice-free, or there is floating ice; for $O = 0$, the ice is grounded either below or above sea level, or the land is ice-free. Using a *continent function* defined as $C(\omega,t) = 1 - O$ is sometimes useful. Since $< c >^o \equiv c$, solving Equation (10) with respect to the FC76 constant gives

$$c(t) = \mathcal{S}^{ave} - < \mathcal{R} >^o, \tag{14}$$

where we have defined $\mathcal{S}^{ave} \equiv < \mathcal{S} >^o$. Hence, using Equation (14) into (10), the SLE is further transformed into

$$\mathcal{S}(\omega,t) = \mathcal{R} + \mathcal{S}^{ave} - < \mathcal{R} >^o. \tag{15}$$

The response function $\mathcal{R}(\omega, t)$ embodies all the interactions qualitatively described in Figure 2; following SM19, we split it into a contribution due to surface loads *and* gravitational attraction (labeled by *sur*) and one due to rotational effects (*rot*), with

$$\mathcal{R}(\omega, t) = \mathcal{R}^{sur} + \mathcal{R}^{rot}, \tag{16}$$

where $\mathcal{R}^{sur}(\omega, t) = \mathcal{G}^{sur} - \mathcal{U}^{sur}$ and $\mathcal{R}^{rot}(\omega, t) = \mathcal{G}^{rot} - \mathcal{U}^{rot}$. According to Farrell [35], $\mathcal{R}^{sur}$ is given by a 3-D spatio-temporal convolution that involves the *surface Green's function for sea level* $\Gamma^s$ and the *surface load variation* $\mathcal{L} = L - L_0$, where $L$ and $L_0$ denote the surface load in an arbitrary state and in the reference state, respectively, and

$$\mathcal{R}^{sur}(\omega, t) \equiv \Gamma^s \otimes \mathcal{L}, \tag{17}$$

while following Milne and Mitrovica [36], $\mathcal{R}^{rot}$ can expressed as a 1-D time convolution between the *rotation Green's function for sea level* $Y_l^s$ and the *centrifugal potential variation*, with

$$\mathcal{R}_{lm}^{rot}(t) = Y_l^s * \Lambda_{lm}, \tag{18}$$

where $(l, m)$ are the spherical harmonic degree and order, respectively (with $l = 0, 1, 2, \ldots, l_{max}$ and $|m| \leq l$). Han and Wahr [37] and Milne and Mitrovica [36], however, have shown that $\Lambda(\gamma, t)$ is essentially a spherical harmonic function of degree and order $(l, m) = (2, \pm 1)$. The Green's functions $\Gamma^s$ and $Y_l^s$ are expressed by particular combinations of *loading Love numbers* and *tidal Love numbers*, respectively. It is important to note that the harmonic coefficients of $\mathcal{R}^{sur}(\omega, t)$, i.e., $\mathcal{R}_{lm}^{sur}(t)$, depend linearly from those of the surface load variation $\mathcal{L}_{lm}(t)$ (see supplement of SM19 for details).

An explicit expression for $\mathcal{S}^{ave}$ in Equation (15) is obtained applying the mass conservation principle that according to SM19 can be stated in various equivalent ways. Here it is convenient to use the form

$$< \mathcal{L} >^e (t) = 0, \tag{19}$$

where the average over the whole Earth's surface is defined, in analogy with Equation (12), as $< \cdots >^e (t) \equiv (1/A^e) \int_e (\cdots) \, dA$. We refer to mass conserving loads that obey Equation (19) as *physically plausible* loads. As shown in SM19, condition (19) is equivalent to

$$\mathcal{L}_{00}(t) = 0, \tag{20}$$

where $\mathcal{L}_{00}(t)$ is the spherical harmonic component of the surface load for degree and order $(l, m) = (0, 0)$. Using the result

$$L(\omega, t) = \rho^i IC + \rho^w BO, \tag{21}$$

(SM19) and some algebra, from the constraint of mass conservation we obtain

$$\mathcal{S}^{ave}(t) = \mathcal{S}^{equ} + \mathcal{S}^{ofu}, \tag{22}$$

where $\mathcal{S}^{equ}$ (equivalent sea-level change) is defined as

$$\mathcal{S}^{equ}(t) = -\frac{\mu}{\rho^w A^o}, \tag{23}$$

with $\mu(t) = \rho^i \int_e (IC - I_0 C_0) \, dA$ denoting the time variation of the grounded ice mass, and term

$$\mathcal{S}^{ofu}(t) = \frac{1}{A^o} \int_e T_0 (O - O_0) \, dA, \tag{24}$$

is associated with ocean function variations, where $T_0$ and $O_0$ are the initial topography and the initial OF, respectively. We note that in the fixed-shorelines approximation of FC76, the OF is constant, with $O = O_0 = O^p$ where $O^p$ is the present OF. Hence, in this approximation $\mathcal{S}^{ofu} = 0$, and $\mathcal{S}^{equ}$ is equivalent to what in the geological literature is often called *eustatic* [9] sea-level change

$$\mathcal{S}^{eus}(t) = -\frac{\mu}{\rho^w A^{op}}, \tag{25}$$

where $A^{op} = \int_e O^p \, dA$ is the present-day area of the oceans.

The SLE (15), complemented by Equations (16)–(18) and (22) constitutes a 3-D non-linear integral equation in the unknown $\mathcal{S}(\omega, t)$, somewhat similar to a 1-D non-homogeneous Fredholm equation of the second kind (see e.g., [38]). Assuming fixed shorelines, as in FC76, would reduce the SLE to a linear equation [31]. The integral, or implicit, nature of the SLE becomes apparent when it is recognized that the response function $\mathcal{R}$ functionally depends, through $\mathcal{G}$ and $\mathcal{U}$, upon $\mathcal{S}$ itself (see SM19). In modern approaches to GIA, the SLE is solved recursively in the spectral domain, adopting the *pseudo-spectral* method [39,40].

In the general case given by Equation (15), no analytical solutions exist for the SLE. However, a closed-form solution can be found in the eustatic approximation, expressed by (25), valid in the very special case of a rigid Earth in which the gravitational attraction between the three components of the SLE is neglected (see Figure 2). Another analytical solution is found assuming a rigid Earth and uniform oceans but allowing for the gravitational interaction between the ice sheets and the oceans, i.e., neglecting the self-attraction of oceans. This solution, often referred to as *Woodward solution* [6], has been discussed in detail by e.g., Spada [31]. Although oversimplified, it is historically important and it has the merit of demostrating the important role of gravitational attraction in shaping the sea surface, with a sea-level change departing from the spatially uniform eustatic solution both nearby the melting ice sheets and in their far field.

## 3. Methods

Spada and Melini [33] have recently released a general open-source Fortran program called SELEN$^4$ (SELEN version 4.0) that solves the SLE in its full form; this shall be employed in next sections to study the geometry of the GIA fingerprints associated with the melting of past ice sheets. SELEN$^4$ is the current stage of the evolution of program `SELEN` which was originally published in 2007 by Spada and Stocchi [41] based upon the theory detailed in Spada and Stocchi [42].

SELEN$^4$ implements the pseudo-spectral method of Mitrovica and Peltier [39] and Mitrovica and Milne [40]. In SELEN$^4$, all the variables have a piecewise constant time evolution. In space, the discretization is performed adopting the equal-area icosahedron-based spherical geodesic grid designed by Tegmark [43], whose density is controlled by the resolution parameter $R$. In our computations, we have set $R = 44$, corresponding to $P = 40R(R-1) + 12 = 75,692$ pixels over the sphere, each having a radius of $\sim$46 km. In this way, the number of cells is comparable to that of a traditional $1° \times 1°$ spherical grid, i.e., 64,800. The spherical harmonic expansions required in the framework of the pseudo-spectral approach are truncated at degree $l_{max} = 128$ and the coefficients are evaluated taking advantage of the quadrature rule for the Tegmark grid [43]. According to SM19, the chosen combination $(R, l_{max})$ ensures a sufficient precision without being computationally too demanding.

In SELEN$^4$, we have implemented the GIA model ICE-5G of Peltier [44]. The ice thickness has been discretised on the Tegmark grid and reduced, at a given pixel, to a uniform sequence of identical time steps with a length of 500 years. The LGM is at 21 ka, and prior to that isostatic equilibrium is assumed. Since the LGM, ICE-5G releases a total equivalent sea level

$$\text{ESL} = \left(\rho^i / \rho^w\right)\left(\Delta V^i / A^{op}\right) = 127.3 \text{ m}, \tag{26}$$

where we have assumed $\rho^i = 931.0$ and $\rho^w = 1000.0$ kg m$^{-3}$, and $\Delta V^i$ is the ice volume variation since the LGM. We combine the ICE-5G deglaciation history with a three-layer volume-averaged version of the VM2 multi-stratified rheological profile [44]. The Maxwell viscosities are $\eta = 2.7, 0.5$ and $0.5$ in units of $10^{21}$ Pa·s in the lower mantle, transition zone and shallow upper mantle, respectively. We assume a fluid inviscid core, a 90 km thick elastic lithosphere and an incompressible rheology. A PREM-averaged [45] density and rigidity profile has been adopted, using a 9-layer radial structure (see Table 1). Loading and tidal Love numbers have been computed using program `TABOO` [46] in a multi-precision environment [47], and expressed in a geocentric reference frame with origin in the center of mass of the whole Earth, including the solid and the fluid portions. To facilitate reproducibility, some values of the Love numbers are listed in Table 2.

**Table 1.** Values of density, rigidity and viscosity adopted in our realization of the rheological model VM2, with abbreviations LT, UM, TZ and LM denoting the lithosphere, the upper mantle, the transition zone and the lower mantle, respectively. With $r_-$ and $r_+$ we denote the radii of the base and of the top of each layer. A few spectral properties of this model are shown in Table 2.

| Radius, $r_-$ (km) | Radius, $r_+$ (km) | Density, $\rho$ (kg m$^{-3}$) | Rigidity, $\mu$ Pa $\times 10^{11}$) | Viscosity, $\eta$ (Pa·s $\times 10^{21}$) | Layer |
|---|---|---|---|---|---|
| 6281.000 | 6371.000 | 3192.800 | 0.596 | $\infty$ | LT |
| 6151.000 | 6281.000 | 3369.058 | 0.667 | 0.5 | UM1 |
| 5971.000 | 6151.000 | 3475.581 | 0.764 | 0.5 | UM2 |
| 5701.000 | 5971.000 | 3857.754 | 1.064 | 0.5 | TZ1 |
| 5401.000 | 5701.000 | 4446.251 | 1.702 | 2.7 | LM1 |
| 5072.933 | 5401.000 | 4615.829 | 1.912 | 2.7 | LM2 |
| 4716.800 | 5072.933 | 4813.845 | 2.124 | 2.7 | LM3 |
| 4332.600 | 4716.800 | 4997.859 | 2.325 | 2.7 | LM4 |
| 3920.333 | 4332.600 | 5202.004 | 2.554 | 2.7 | LM5 |
| 3480.000 | 3920.333 | 5408.573 | 2.794 | 2.7 | LM6 |
| 0 | 3480.000 | 10931.731 | 0 | 0 | Core |

**Table 2.** Numerical values of the elastic (with superscript $e$) and fluid (with superscript $f$) loading Love numbers $k_l^L$ and $h_l^L$ for the rheological model VM2 (see Table 1), for some harmonic degrees $l$ in the range $1 \le l \le 1024$. We use the compact notation $v_{exp} = v \times 10^{-exp}$ and $v^{exp} = v \times 10^{exp}$, where $v$ is any value in the table and $exp$ is an exponent. Note that, for this model, the elastic tidal Love numbers of degree $l = 2$ are $(k_2^{Te}, h_2^{Te}) = (0.289^0, 0.524^0)$ while the fluid values are $(k_2^{Tf}, h_2^{Tf}) = (0.931^0, 0.191^1)$.

| | $l = 1$ | 2 | 4 | 16 | 64 | 128 | 256 | 512 | 1024 |
|---|---|---|---|---|---|---|---|---|---|
| $k_l^{Le}$ | $-1.000^0$ | $-0.235^0$ | $-0.117^0$ | $-0.558_1$ | $-0.231_1$ | $-0.123_1$ | $-0.642_2$ | $-0.323_2$ | $-0.162_2$ |
| $h_l^{Le}$ | $-0.102_1$ | $-0.442^0$ | $-0.463^0$ | $-0.975^0$ | $-0.165_1$ | $-0.181_1$ | $-0.188_1$ | $-0.190_1$ | $-0.190_1$ |
| $k_l^{Lf}$ | $-1.000^0$ | $-0.980^0$ | $-0.981^0$ | $-0.956^0$ | $-0.192^0$ | $-0.249_1$ | $-0.675_2$ | $-0.323_2$ | $-0.162_2$ |
| $h_l^{Lf}$ | $-0.163_1$ | $-0.267_1$ | $-0.481_1$ | $-0.173_2$ | $-0.139_2$ | $-0.363_1$ | $-0.198_1$ | $-0.190_1$ | $-0.190_1$ |

The SLE has been solved iteratively [48] adopting three "external" iterations to progressively refine the OF and the paleo-topography and, for each of them, performing three "internal" iterations to solve for $\mathcal{S}(\omega, t)$, for a given approximation of topography. According to SM19 and to independent results by Milne and Mitrovica [36], these choices ensure sufficiently precise results. The present-day relief, obtained by a pixelization of the ice-free version of model ETOPO1 [49,50], has been imposed as a *final* condition. The present-day ice distribution is given by the last time step of ICE-5G. Finally, to model the effects of polar motion on sea-level change, we have employed the *revised rotation theory* by Mitrovica et al. [51] and Mitrovica and Wahr [52]. The revised theory accounts, in the polar motion equations [4], for the long-term viscous behavior of the lithosphere and for the effects of internal dynamics on the Earth flattening. Some runs, however, have been performed adopting the *traditional rotation theory* (see e.g., Spada et al. [46] and references therein), which assumes an elastic lithosphere

in the polar motion equations and does not take internal dynamics into account. In some other runs, we are totally neglecting the effects of Earth rotation.

## 4. Some Properties of the GIA Fingerprints

In the next subsections we provide an overview of the properties of the GIA fingerprints for the present-day trends of (i) relative sea-level ($\dot{\mathcal{S}}$), (ii) vertical displacement ($\dot{\mathcal{U}}$), (iii) geoid height ($\dot{\mathcal{G}}$), (iv) absolute sea level ($\dot{\mathcal{N}}$), and (v) surface load ($\dot{\mathcal{L}}$), respectively. The list is by no means exhaustive, and it should be also extended to other quantities associated with GIA, as for example the horizontal displacement and the free air gravity anomalies. Future releases of SELEN[4] shall include modules for these and possibly other GIA fingerprints. Note that since the equipotential surfaces of the gravity field and the solid surface of the Earth are defined at all grid points, the map of $\dot{\mathcal{S}}$ and of all the other fingerprints considered in the following are also extended across the continents. As GIA evolves over millennia, the geometry of the fingerprints would not change appreciably on time scales of a few centuries [53].

### 4.1. Relative Sea-Level Change

Figure 3 shows the GIA fingerprint $\dot{\mathcal{S}}(\omega)$, i.e., the rate of present-day relative sea-level change. Assuming that GIA from the melting of past ice sheets is the unique cause of contemporary sea-level change, the rates shown in Figure 3 would be directly observable as constant secular trends at tide gauges (see e.g., [31]). The $\dot{\mathcal{S}}$ fingerprint shows the major features and patterns of regional variability already described by Mitrovica and Milne [12], i.e., the strong relative sea-level fall associated with post glacial rebound across the polar regions that where once covered by thick ice sheets and corresponding to RSL zone *I* of Clark et al. [2], the sea-level rise across the ring-shaped collapsing lateral fore-bulges (zone *II*), and the region of broad sea-level fall associated with equatorial ocean syphoning (zone *V*). The offshore sea-level rise clearly evident in the equatorial regions in the GIA maps of Mitrovica and Milne [12] and Melini and Spada [32], and linked to continental levering (zone *VI*), does not stand out clearly in Figure 3, except perhaps along the coasts of central Africa and Australia. In part, this could be due to the different deglaciation chronology and rheology adopted in [12,32], corresponding to model ICE-3G (VM1) of Tushingham and Peltier [54]. However, by a further SELEN[4] run, in which we have still adopted model ICE-5G (VM2) but we have ignored rotational effects as done in [12,32], we have ascertained that the localised offshore sea-level rise is clearly detectable. Thus, we conclude that in Figure 3 this feature is blurred by the long-wavelength effects of Earth rotation. The rotational feedback on sea level is responsible for the southern hemisphere swaths of sea-level rise and fall around Oceania and South America, respectively. In the northern hemisphere, these effects are less evident, due to the dominating contribution of glacial unloading and of the peripheral subsidence.

A crude but useful way to simplify the evident geometrical complexity of GIA fingerprint in Figure 3 is to evaluate its spatial average (all spatial averages of the GIA fingerprints discussed in the following are collected in Table 3). For the ocean-average of $\dot{\mathcal{S}}(\omega)$, given by $<\dot{\mathcal{S}}>^o$, the SLE theory provides an explicit formula, which stems from the constraint of mass conservation. It can be obtained by computing the time-derivative of Equation (22), also taking (23) and (24) into consideration. However, a numerical evaluation on the grid is more convenient, which according to our computations gives the small value

$$<\dot{\mathcal{S}}>^o = -0.06 \quad \text{mm year}^{-1}, \tag{27}$$

where conventionally we shall use the term *small* to indicate all GIA rates $< 0.1$ mm year$^{-1}$ in modulus. A two-fold larger but coherent value, with $<\dot{\mathcal{S}}>^o = -0.14$ mm year$^{-1}$, was computed by Spada [31], who adopted the traditional rotation theory and a coarse three-layer rheological profile for VM2. By inspection of Equations (23) and (24), the small value of $<\dot{\mathcal{S}}>^o$ may reflect minor variations of the OF associated to changes in the area $A^o(t)$ of the ocean basins, tiny values of the rate of change of the

grounded ice mass $\mu(t)$, or both. Since in model ICE-5G (VM2) the mass distribution over Greenland has seen small but significant variations during the last $\approx6000$ years that continue to present [55], the average $<\dot{\mathcal{S}}>^o$ effectively reflects both contributions. However, if we had employed a GIA model that assumes no ice sheets fluctuations during the last few kyrs, like ICE-3G (VM1) [54,56] or ICE-6G (VM5a) [57], also imposing fixed shorelines as in FC76, we would have obtained *exactly*

$$<\dot{\mathcal{S}}>^o_{FC76}= 0 \quad \text{mm year}^{-1}, \tag{28}$$

as a direct consequence of mass conservation. In fact, this result would be achieved regardless the rheological profile chosen. We note that the SLE theory tells nothing about the whole-Earth-surface average $<\dot{\mathcal{S}}>^e$, which however according to our computations in Table 3 is not small.

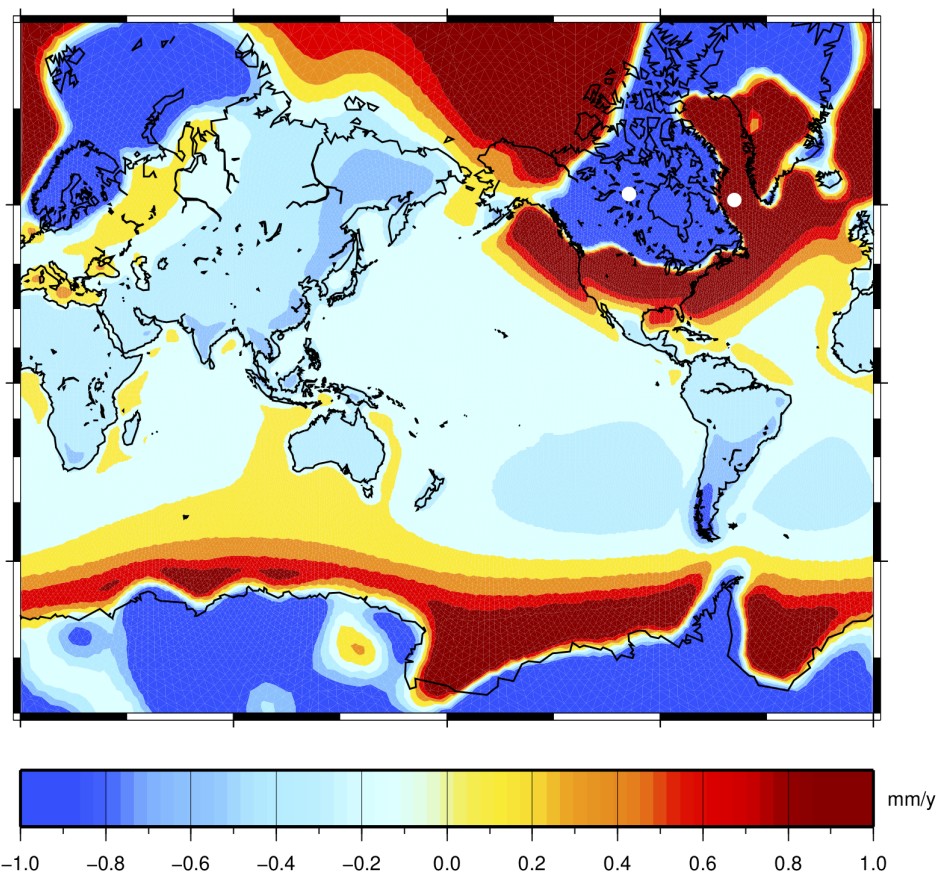

**Figure 3.** GIA fingerprint for $\dot{\mathcal{S}}$, the present-day rate of relative sea-level change, obtained by implementing model ICE-5G (VM2) in SELEN[4]. To better visualize the regional variations, the palette is limited to the range of $\pm1$ mm year$^{-1}$. The largest rates, marked by white dots, are associated with the isostatic disequilibrium still caused by the disintegration of the Laurentide ice sheet complex, with $\dot{\mathcal{S}} \sim -17.9$ and $\dot{\mathcal{S}} \sim +5.3$ mm year$^{-1}$, respectively. The most significant regional variability, measured as the density of local maxima and minima of $\dot{\mathcal{S}}$ in this map, is found to within $\sim1500$ km from the continental margins.

**Table 3.** Ocean (top) and whole-Earth surface averages (bottom) of the present-day rate of change of GIA fingerprints considered in this study. In this table, the outputs of SELEN$^4$ have been rounded to two significant figures. Although in the text we dwelt upon the new rotation theory (column (a)), results for the the traditional theory are also shown here in (b) while in (c) no rotational effects are taken into account. It is apparent that the spatial averages are only moderately affected by the choice of the rotation theory. The values of $< \dot{\mathcal{U}} >^e$, $< \dot{\mathcal{G}} >^e$ and $< \dot{\mathcal{L}} >^e$ are numerically found to be $<10^{-5}$ mm year$^{-1}$ in modulus. By virtue of mass conservation, their expected theoretical value should be exactly zero.

| Average | (a) New Theory (mm year$^{-1}$) | (b) Traditional Theory (mm year$^{-1}$) | (c) No Rotation (mm year$^{-1}$) |
|---|---|---|---|
| $< \dot{\mathcal{S}} >^o$ | −0.06 | −0.06 | −0.06 |
| $< \dot{\mathcal{U}} >^o$ | −0.27 | −0.30 | −0.24 |
| $< \dot{\mathcal{N}} >^o$ | −0.33 | −0.35 | −0.30 |
| $< \dot{\mathcal{G}} >^o$ | −0.06 | −0.09 | −0.04 |
| $< \dot{\mathcal{L}} >^e$ | +0.00 | +0.00 | +0.00 |
| $< \dot{\mathcal{S}} >^e$ | −0.27 | −0.27 | −0.26 |
| $< \dot{\mathcal{U}} >^e$ | +0.00 | +0.00 | +0.00 |
| $< \dot{\mathcal{N}} >^e$ | −0.27 | −0.27 | −0.26 |
| $< \dot{\mathcal{G}} >^e$ | +0.00 | +0.00 | +0.00 |

*4.2. Vertical Displacement*

In Figure 4 we show the GIA fingerprint $\dot{\mathcal{U}}(\omega)$, which represents the present-day rate of change of the vertical displacement that would be observed, at a given location, by an earthbound GPS receiver [58–61]. By a visual inspection, it is apparent that most of the features of this map are anti-correlated with those shown by fingerprint $\dot{\mathcal{S}}(\omega)$ in Figure 3. In particular, this occurs in previously glaciated areas and in their surroundings, where a relative sea-level rise is accompanied by subsidence, and *viceversa*. However, we note that apparently paradoxical conditions as having a relative sea-level rise in uplifting regions, or a relative sea-level fall in subsiding regions, are not forbidden *a priori* by the SLE (see Equation (5)). These conditions may well occur where the rate of absolute change $\dot{\mathcal{N}}(\omega)$, considered below, attains positive and negative values, respectively.

The anti-correlation between $\dot{\mathcal{U}}(\omega)$ and $\dot{\mathcal{S}}(\omega)$ is not so evident across the equatorial basins, where the $\dot{\mathcal{U}}(\omega)$ fingerprint shows a clear sectorial symmetry of harmonic degree and order $(l, m) = (2, \pm 1)$, which manifests the long-wavelength effects of Earth rotation. By a comparison with Figure 3, it turns out that such symmetry is definitively more compelling for $\dot{\mathcal{U}}(\omega)$ than for $\dot{\mathcal{S}}(\omega)$. In the northern hemisphere, the rotation-induced subsidence across North America counteracts the uplift associated with the melting of Laurentide ice sheet, but it intensifies the subsidence across the peripheral fore-bulges. Conversely, in Asia the effects associated to Earth rotation are clearly enhancing the vigor of the uplift induced by continental levering [12]. Interestingly, Figure 4 reveals that a number of GIA-associated processes coherently concur to the uplift in Patagonia, which is caused by local effects due to un-loading of the former Patagonian ice sheet included in model ICE-5G (VM2), by the contribution of continental levering and by the effect from Earth rotation. The unloading associated with the melting of contemporary glaciers and ice caps [62,63], which however is not taken into account in our modeling, would act is the same direction.

As we have done for $\dot{\mathcal{S}}(\omega)$ above, it is useful now to consider spatial averages of the fingerprint in Figure 4. To a very high precision (see Table 3), the whole-Earth average of $\dot{\mathcal{U}}(\omega)$ is numerically found to be

$$< \dot{\mathcal{U}} >^e = 0.00 \quad \text{mm year}^{-1}, \tag{29}$$

a property of the $\dot{\mathcal{U}}(\omega)$ fingerprint that, once again, is explained in terms of the principle of mass conservation. Since we have assumed a *plausible* surface load, mass conservation is ensured by Equation (20). From Equation (17), this implies a vanishing $\mathcal{R}^{sur}(\omega, t)$ at harmonic degree and order

$(l, m) = (0, 0)$, from which the fundamental property (29) of the $\dot{\mathcal{U}}(\omega)$ fingerprint follows immediately. We note that this characteristic is totally unaffected by the choice of the GIA model and, in particular, from the Earth rheological profile assumed. It also holds true when, in GIA modeling, one neglects rotational effects and the horizontal migration of the shorelines, as done for example in the FC76 formulation (this is confirmed by the results in Table 3). Furthermore, as long as the mass conservation constraint is not violated, it is also valid for the $\dot{\mathcal{U}}(\omega)$ fingerprint associated to the present melting of continental ice sheets, for which viscous rheological effects can be neglected [31,34].

## U–dot for run I5G/R44/L128/I33 NT

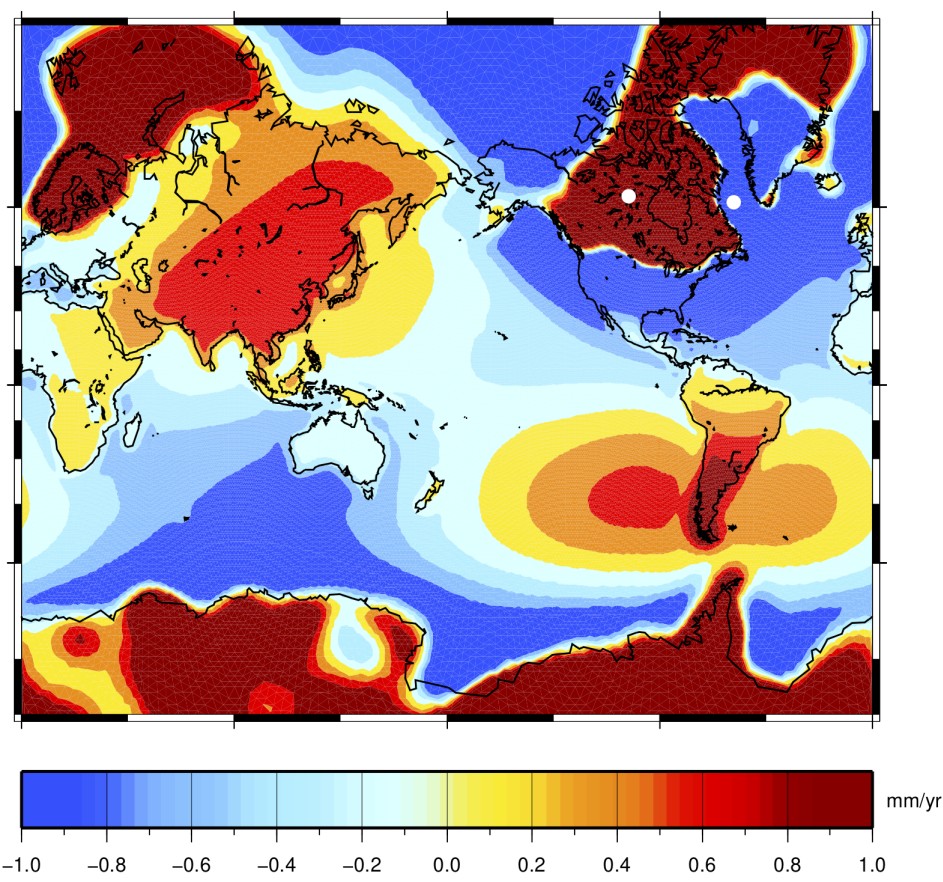

**Figure 4.** GIA fingerprint for the current rate of crustal uplift $\dot{\mathcal{U}}$, according to our implementation of GIA model ICE-5G (VM2). The rates with largest absolute values, marked by white dots, are associated with the melting of the Laurentide ice sheet in north America and Canada, and are found in the same locations of Figure 3, with values of $\dot{\mathcal{U}} \sim +19.2$ and $\dot{\mathcal{U}} \sim -5.7$ mm year$^{-1}$, respectively. The regional variability of the $\dot{\mathcal{U}}$ fingerprint appears to be comparable to that of $\dot{\mathcal{S}}$ in Figure 3 but the rotational lobes are much more developed.

We finally note that the SLE theory tells nothing about the GIA-induced average rate of subsidence of the ocean floors $< \dot{\mathcal{U}} >^o$, which however according to our computations reported in Table 3, is found not to be small. This would support the idea of a significant influence of climate variations on the isostatic equilibrium of the sea floor topography [64,65]. The negative value of $< \dot{\mathcal{U}} >^o$ is easily justified by the dominance, in Figure 4, of blue swaths across the oceans caused by the effect of water loading. Conversely, by the argument of mass conservation, we expect a not small and positive value $< \dot{\mathcal{U}} >^c$, where superscript $c$ denotes the average over the continents. We shall return on this issue in Section 5 below.

### 4.3. Geoid Height and Absolute Sea-Level Change

In Figure 5 we show the map of the GIA fingerprint for $\dot{\mathcal{G}}(\omega)$. According to Equation (9), this quantity represents the present-day rate of change of the geoid height. It appears that $\dot{\mathcal{G}}(\omega)$ is characterized by a well developed lobed symmetry with $(l, m) = (2, \pm 1)$ and, with respect to $\dot{\mathcal{S}}(\omega)$ and $\dot{\mathcal{U}}(\omega)$, by an overall smoother resemblance. The cause is to be found in the different spectral content of the $h_l^L(t)$ and $k_l^L(t)$ loading Love numbers that contribute to $\dot{\mathcal{U}}(\omega)$ and $\dot{\mathcal{G}}(\omega)$, respectively; see SM19 for details. The pattern associated to Earth rotation is so strong that the regional effects from glacial unloading are only just visible in the polar regions of both hemispheres. To suitably interpret the $(l, m) = (2, \pm 1)$ symmetry, it is worth to note that according to our computations, the GIA-induced polar motion presently occurs at a rate of $\sim$1.4 deg/Myr (roughly corresponding to 15 cm/year on the Earth's surface) along the meridian $\sim$80° W (roughly, towards the Hudson Bay). Such rate and direction of polar drift match well the astronomical observations in the course of last century (see e.g., Lambeck [4]) and with recent analyses about the causes of secular polar motion [66]. Performing a further run of SELEN$^4$ in which we have adopted the *traditional rotation theory* (see e.g., Spada et al. [46]), we have verified that the $(l, m) = (2, \pm 1)$ pattern of $\dot{\mathcal{G}}(\omega)$ would be indeed much stronger, with a more than two-fold rate of polar drift of $\sim$3.5 deg/Myr in the same direction. The enhanced rate of polar motion implied by the traditional rotation theory compared to the new theory is in full agreement with the analysis of Mitrovica et al. [51] and Mitrovica and Wahr [52].

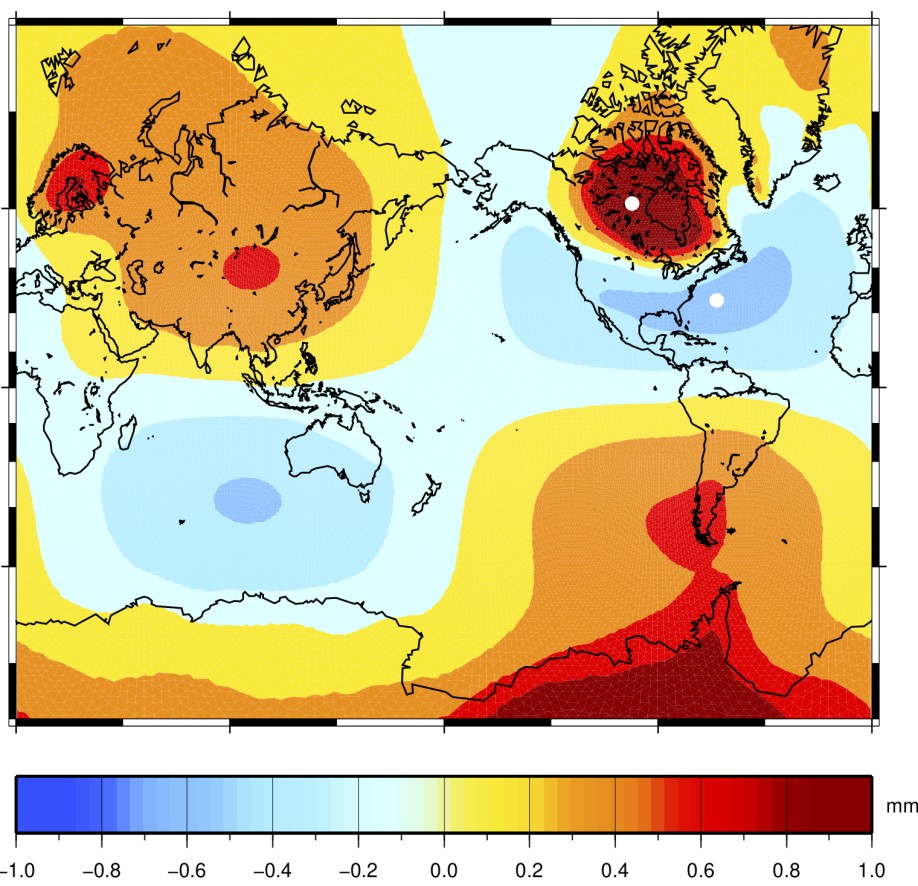

**Figure 5.** GIA fingerprint for $\dot{\mathcal{G}}$ i.e., the current rate of geoid height variation, according to our GIA simulation based upon model ICE-5G (VM2). The white dots show where the largest rates are predicted, with values of $\dot{\mathcal{G}} \sim -0.6$ and $\dot{\mathcal{G}} \sim +1.7$ mm year$^{-1}$, respectively. The regional variability in this map, i.e., the alternation of local minima and maxima, is drastically reduced in comparison with $\dot{\mathcal{S}}$ and $\dot{\mathcal{U}}$, giving to $\dot{\mathcal{G}}$ a very smooth semblance.

Based upon the same argument we have used for $\dot{\mathcal{U}}(\omega)$ above (i.e., mass conservation ensured by plausible surface loads), the fundamental property of the $\dot{\mathcal{G}}(\omega)$ fingerprint can be similarly expressed by

$$< \dot{\mathcal{G}} >^e = 0.00 \quad \text{mm year}^{-1}, \tag{30}$$

which we have verified numerically to be valid to a very high precision (see Table 3). In consequence of (30), harmonics with $(l, m) = (0,0)$ are not contributing to $\dot{\mathcal{G}}(\omega)$. We further note that condition $< \dot{\mathcal{G}} >^o \approx < \dot{\mathcal{S}} >^o$, suggested by the results in column (a) of Table 3, is due to chance and it is not reflecting any particular property of the GIA fingerprints. Indeed, when the traditional rotation theory is adopted or rotation is neglected, as done in columns (b) and (c), respectively, or alternative GIA models such as ICE-6G (VM5a) are employed as in SM19, this condition is not met.

As shown by e.g., Melini and Spada [32], the individual harmonic components of $\dot{\mathcal{G}}(\omega)$, i.e., $\dot{\mathcal{G}}_{lm}$, are proportional to the rates of change of the GIA-induced variations of the Stokes coefficients of the Earth's gravity field, detectable by the Gravity Recovery and Climate Experiment (GRACE); see Wahr et al. [67] for a discussion. In particular,

$$\overline{\dot{\delta c}}_{lm} + i\,\overline{\dot{\delta s}}_{lm} = a^{-1}\,\sqrt{2 - \delta_{0m}}\,\dot{\mathcal{G}}^*_{lm}, \tag{31}$$

where $\overline{\delta c}_{lm}$ and $\overline{\delta s}_{lm}$ are the GIA-induced variations of the fully normalised cosine and sine Stokes coefficients, $i = \sqrt{-1}$ is the imaginary unit, $a$ is the reference Earth's radius, $\delta_{ij}$ is the Kronecker delta, and the asterisk denotes complex conjugation. We also note that since we are solving the SLE in a geocentric reference frame with origin in the whole-Earth center of mass, a further property of the field $\dot{\mathcal{G}}(\omega)$ is that of not including contributions from the harmonics of degree and order $(l, m) = (1, 0)$ and $(l, m) = (1, \pm 1)$ [68]. Hence, in Equation (31), only terms with harmonic degree $l \geq 2$ appear.

The GIA fingerprint for $\dot{\mathcal{N}}(\omega)$, shown in Figure 6, represents the present-day rate of change of the sea surface height (or absolute sea level) that would be observed across the oceans by satellite altimetry [22,31], assuming that only GIA is contributing to contemporary sea-level change. It is worth to recall that, regardless the rotation theory adopted in GIA modeling, the $\dot{\mathcal{N}}(\omega)$ fingerprint is not independent upon $\dot{\mathcal{S}}(\omega)$ and $\dot{\mathcal{U}}(\omega)$, since from the basic form of the SLE (see Equation (5)), we have $\dot{\mathcal{N}}(\omega) = \dot{\mathcal{S}}(\omega) + \dot{\mathcal{U}}(\omega)$. Actually, in view of the relatively small range of values spanned by $\dot{\mathcal{N}}(\omega)$, which never exceeds the value of 1.5 mm year$^{-1}$ in modulus, the approximation of the SLE $\dot{\mathcal{S}}(\omega) \approx -\dot{\mathcal{U}}(\omega)$ is inviting, but it would be an oversimplification. We further note that only in the idealized case of an un-deformable Earth, with $\dot{\mathcal{U}}(\omega) = 0$, absolute and relative sea-level variations would coincide, with $\dot{\mathcal{S}}(\omega) = \dot{\mathcal{N}}(\omega)$.

By the FC76 formula, it turns out that $\dot{\mathcal{N}}(\omega)$ (see Equation (6) and Figure 1) is strongly associated to the rate of geoid change, since it simply differs from $\dot{\mathcal{G}}(\omega)$ by the spatially invariant quantity $\dot{c}$, where $c$ is the FC76 constant. In consequence of the FC76 formula, the whole-Earth surface average of $\dot{\mathcal{N}}(\omega)$ is

$$< \dot{\mathcal{N}} >^e = \dot{c} = -0.27 \quad \text{mm year}^{-1}, \tag{32}$$

which turns out to be an appealingly simple definition of $\dot{c}$. Using the gridded data shown in Figure 6, we numerically obtain a not small ocean average

$$< \dot{\mathcal{N}} >^o = -0.33 \quad \text{mm year}^{-1}, \tag{33}$$

a (GIA model dependent) value closely matching the value of $-0.3$ mm year$^{-1}$, often adopted as a *rule of thumb* to correct the altimetric absolute sea-level trend for the effects of past GIA (see [31,34] and references therein). Since during the altimetry era (1992-today) the rate of global mean sea-level rise has well exceeded $\sim$3 mm year$^{-1}$ [69,70], using the average (33) to perform the GIA correction is certainly justified. However, spatial trends of $\dot{\mathcal{N}}(\omega)$ at a regional scale may become important when one considers the effects of present land ice on absolute sea level change, as done by Ponte et al. [71].

## N–dot for run I5G/R44/L128/l33 NT

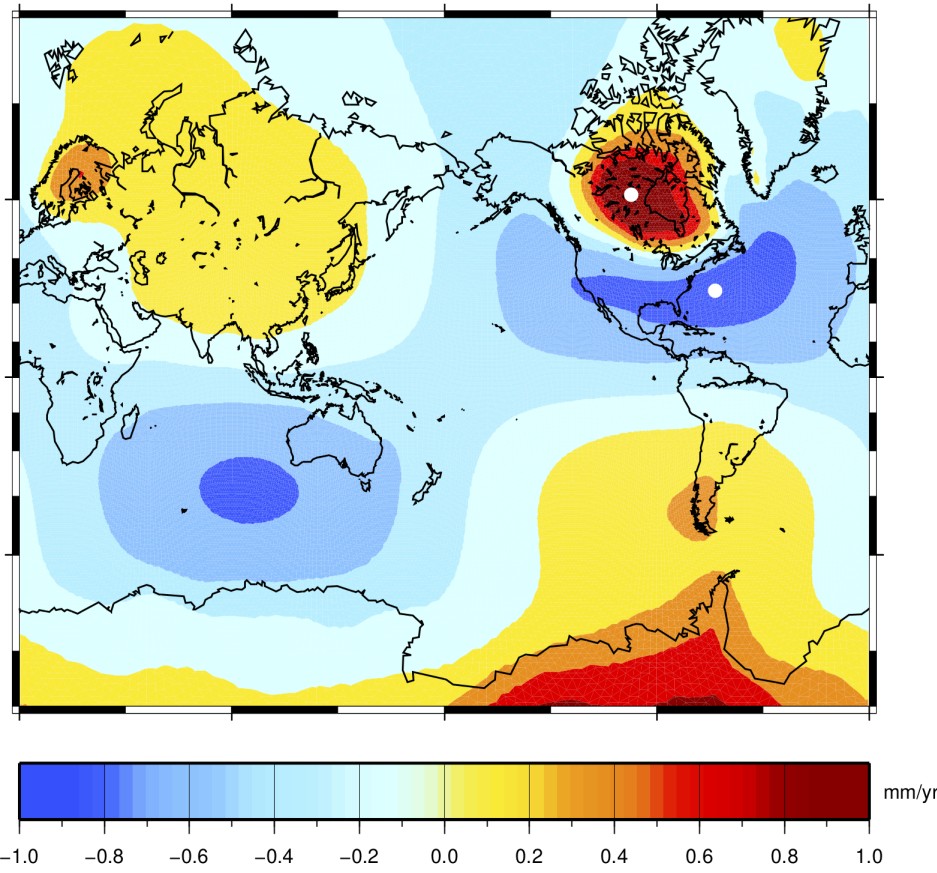

**Figure 6.** Fingerprint for $\dot{\mathcal{N}}$, which represents the current rate of sea surface variation or absolute sea-level change according to our implementation of GIA model ICE-5G (VM2). White dots mark the places where the largest rates are expected, with $\dot{\mathcal{N}} \sim -0.9$ and $\dot{\mathcal{N}} \sim +1.4$ mm year$^{-1}$, respectively. The spatial variability of $\dot{\mathcal{N}}$ matches that of $\dot{\mathcal{G}}$ in Figure 5, since the two fingerprints only differ by the spatially invariant term $\dot{c}$, where $c$ is the FC76 constant.

### 4.4. Surface Load

We conclude our overview with a few remarks about the GIA fingerprint for $\dot{\mathcal{L}}(\omega)$, the present-day rate of change of the surface load. This quantity, which is shown in Figure 7 in units of mm year$^{-1}$ of water equivalent, describes the local variations in the distribution of the ice and water. We recall that the load variation $\mathcal{L}(\omega, t)$ is defined as $L - L_0$, where $L(\omega, t)$ is given by Equation (21) and $L_0$ is the value of $L$ in the reference state (see Figure 1). To interpret the gross features of the map shown in Figure 7, for one moment it is convenient to assume that the continent function $C$ and the ocean function $O$ are constant to the present day values, as it would be implicit in the FC76 formulation of GIA. If this holds true, by evaluating the time-derivative of Equation (21) at present time we obtain

$$\dot{\mathcal{L}}(\omega) \simeq \rho^i \dot{\mathcal{I}} C + \rho^w \dot{\mathcal{S}} O, \tag{34}$$

where $\mathcal{I} = I - I_0$ is the ice thickness variation and we have also used the definition of sea-level change given by Equation (4). Across the oceans, the existence of the positive correlation between $\dot{\mathcal{L}}(\omega)$ and $\dot{\mathcal{S}}(\omega)$ predicted by Equation (34) is easily recognized comparing the fingerprints in Figures 3 and 7. The strong contribution to $\dot{\mathcal{L}}(\omega)$ across Greenland is associated with the current ice variation that ICE-5G (VM2) embodies in this region [44,55]; in all other continental areas the load variation vanishes, in agreement to Equation (34). A notable exception is West Antarctica, where the negative trend of the load is associated with the significant variations of the ocean function in this region, associated with

the still continuing transition between grounded and floating ice. However, this is not accounted for in the FC76 approximation (34), which assumes a constant OF. Lastly, we observe that once integrated over the whole Earth's surface, $\mathcal{L}(\omega, t)$ gives the global mass change of the system with respect to the reference state. However, since mass is conserved, consistently with Equation (19), we have

$$< \dot{\mathcal{L}} >^e = 0.00 \quad \text{mm year}^{-1}, \tag{35}$$

which according to Table 3 is numerically verified to a very high precision.

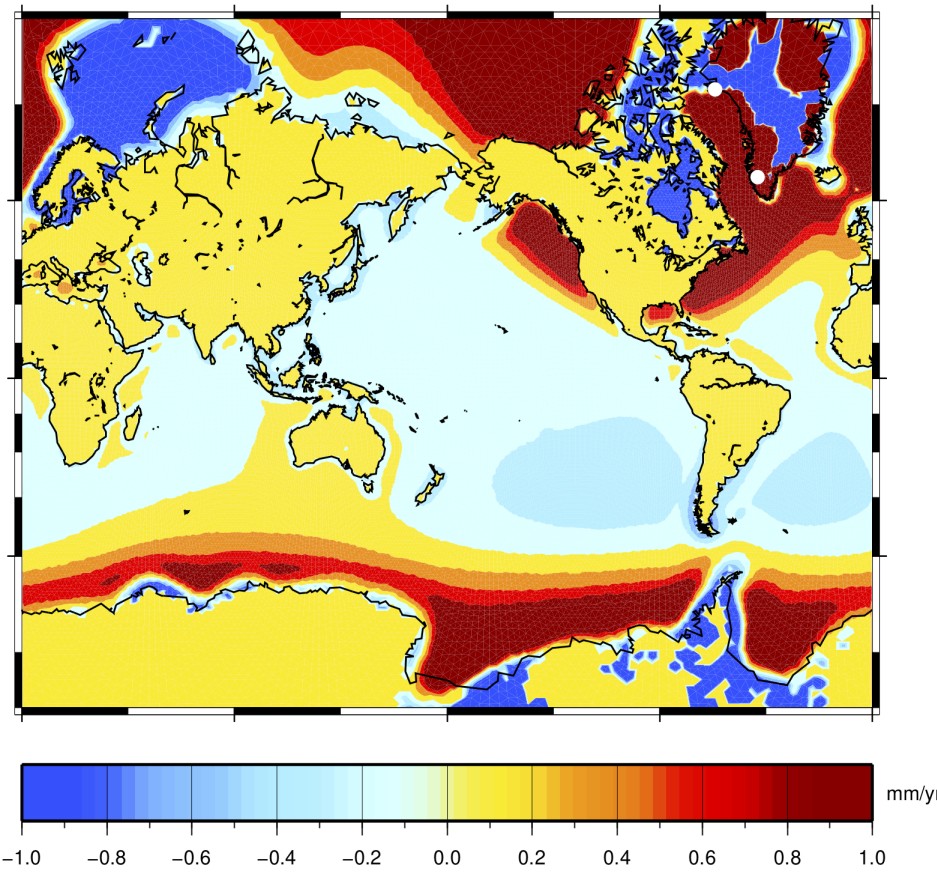

**Figure 7.** GIA fingerprint for the present-day rate of variation of the surface load $\dot{\mathcal{L}}(\omega)$, in units of mm year$^{-1}$ of water equivalent. The whole-Earth average is $< \dot{\mathcal{L}} >^e = 0.00$ mm year$^{-1}$ to a very high precision. In oceanic areas, $\dot{\mathcal{L}}$ is strongly correlated with $\dot{\mathcal{S}}$ in Figure 3. In continental areas, $\dot{\mathcal{L}}$ only takes contributions in regions where, according to ICE-5G (VM2), ice thickness variations are still occurring or where the OF is still varying. These conditions are met in Greenland, where $\dot{\mathcal{L}}$ shows the extreme values (white dots), and in West Antarctica, respectively.

## 5. Observing the Global GIA Fingerprint by Vertical GPS Rates

As an example of application of the fingerprints properties illustrated above, we mention the problem of directly using geodetic observations to quantify the present global pattern of GIA. Recently, using a large global compilation of geodetic GPS rates in conjunction with a Bayesian inference method, Husson et al. [72] have reconstructed and visualized the long-wavelength signature of GIA on the rate of present day vertical crustal uplift. In principle, once the contributions from short-wavelength tectonic phenomena have been filtered out, the geodetically observed rates across the continents should match those predicted by current GIA models, at least in their global traits and in their spatial averages. One possible way to verify this consistency is to consider the average over the continents of the

geodetically determined rate of vertical uplift $< \dot{\mathcal{U}} >^c (t)$, where $< \cdots >^c (t) \equiv (1/A^c) \int_c (\cdots) \, dA$, $A^c(t) = A^e - A^o$ being the area of the continents. In Husson et al. [72], the scalar field $\dot{\mathcal{U}}$ has been estimated from GPS vertical rates by a self-adaptive trans-dimensional regression that exploits the properties of the Voronoi tesselation. The pattern of GIA inferred by regression has been found to broadly resemble the one that we would expect by a model like ICE-5G (VM2), provided that components with wavelengths <2500 km are removed.

Of course, the average $< \dot{\mathcal{U}} >^c$ could be evaluated numerically using the results of Figure 4 and compared to the value obtained from the pattern of the GPS rates. However, through the SLE, it is straightforward and possibly more meaningful to express $< \dot{\mathcal{U}} >^c$ in terms of the ocean-averaged fingerprints $< \dot{\mathcal{S}} >^o$ and $< \dot{\mathcal{N}} >^o$ that we have already discussed above in view of their particular significance. Here we largely follow Husson et al. [72] and, since we simply aim at illustrating the method, we do not consider the modeling uncertainties on the GIA fingerprints. On one hand, taking advantage of Equation (29), we have

$$\frac{1}{A^e} \int_e \dot{\mathcal{U}} \, dA = 0, \tag{36}$$

which by the additivity of the surface average gives

$$\frac{1}{A^e} \left( \int_c \dot{\mathcal{U}} \, dA + \int_o \dot{\mathcal{U}} \, dA \right) = 0, \tag{37}$$

or, equivalently,

$$\frac{A^c}{A^e} < \dot{\mathcal{U}} >^c + \frac{A^o}{A^e} < \dot{\mathcal{U}} >^o = 0, \tag{38}$$

where we have used the definitions of continent and ocean average; hence

$$< \dot{\mathcal{U}} >^c = -\frac{A^o}{A^c} < \dot{\mathcal{U}} >^o . \tag{39}$$

On the other hand, ocean-averaging both sides of the SLE in the form (5) gives

$$< \dot{\mathcal{U}} >^o = < \dot{\mathcal{N}} >^o - < \dot{\mathcal{S}} >^o, \tag{40}$$

which used into (39) yields the average of $\dot{\mathcal{U}}$ across the continents

$$< \dot{\mathcal{U}} >^c = \frac{A^o}{A^c} \left( < \dot{\mathcal{S}} >^o - < \dot{\mathcal{N}} >^o \right), \tag{41}$$

which is only expressed in terms of ocean-averaged fingerprints.

The exercise above shows that the average vertical uplift across the continents determined by GPS could be estimated, *in principle*, by ocean-averaging the tide gauge trends and subtracting the ocean averaged altimetry-derived rate of absolute sea-level change. This would hold true regardless the GIA model employed. By approximating $A^o \approx (7/10)A^e$ and $A^c \approx (3/10)A^e$, so that $A^o/A^c \approx 7/3$, and using the ocean averages for ICE-5G (VM2) given by Equations (27) and (33), we obtain the not small value

$$< \dot{\mathcal{U}} >^c = 0.63 \quad \text{mm year}^{-1}, \tag{42}$$

where we note that since in Equation (41) Husson et al. [72] have used the crude fixed-shorelines approximation that implies $< \dot{\mathcal{S}} >^o = 0$ they have obtained slightly different values for $< \dot{\mathcal{U}} >^c$. Nevertheless, the value of $< \dot{\mathcal{U}} >^c$ matches very well the rate effectively reconstructed by Husson et al. [72] (0.64 mm year$^{-1}$). This suggests that the trans-dimensional regression method has been effective in isolating the fingerprint of GIA. As pointed by Husson et al. [72], the effects of current melting of glaciers and ice caps in response to global warming would not alter substantially our estimate in (42).

## 6. Conclusions

In this work we have reviewed some aspects of GIA, i.e., the response of the Earth to the disequilibrium caused by the melting of the late-Pleistocene ice sheets. Arguments based upon the physical properties of the SLE have been corroborated by results obtained from up-to-date numerical tools in GIA modeling. Among the processes that concur to present sea-level rise, the special role of GIA has been recognized long ago; in fact, only GIA is affected by the rheology of the Earth and, at the same time, it affects significantly the gravity field and the rotational state of the planet. Although, according to GIA models, the deglaciation of the late-Pleistocene ice sheets came to an end thousands of years ago, at present the effects of GIA are still significant and they influence a number of directly observable geophysical and geodetic quantities. Since GIA evolves slowly, its contribution to the instrumental observations will persist also during next centuries although it shall gradually fade away. Model predictions show that the computed patterns or fingerprints of GIA are characterized by an outstanding complexity. In our roundup of the general properties of the GIA fingerprints, we have considered both the geometrical and the physical aspects of such complexity, emphasizing their spatial symmetries and regional character, which we have interpreted qualitatively and quantitatively with the aid of the SLE.

The study of the relative sea level fingerprint has revealed that at present the role of GIA is not that of causing an effective, mean global change. Rather, it causes essentially local regional effects which strongly contaminate the tide gauges records but that *almost* wash out when averaged over the present-day oceans, leaving a small contribution reflecting minor variations in the area of the sea floor and possibly current variations of ice thickness, when these are accounted for by the GIA model. The coastal regions are certainly those being most affected by the regional variability of GIA. The pattern of the GIA-induced vertical uplift is extremely variegated but it globally averages out to zero as an effect of mass conservation. Although it is largely anti-correlated to that of relative sea level, it shows more clearly the mark of the rotational effects of GIA, with a symmetry dominated by a very long-wavelength harmonic pattern. By a specific example from the recent literature, we have shown that the basic traits of the GIA fingerprint of vertical displacement can be visualized using data from a large global compilation of geodetic GPS rates. The symmetry imposed by the polar drift of the rotation axis is even more enhanced when one considers the fingerprints of the geoid height variation and of the absolute sea-level change, which only differ by a spatially invariant term. These two last signatures of GIA have presently a particular role in physical geodesy, since they are commonly employed to purge the trends of the Stokes coefficients of the gravity field and the sea-level altimetric records from the GIA effects.

In this study, we have employed only one of the ICE-*X* models of WR Peltier and collaborators [44], leaving a systematic exploration to a further study. Unquestionably, the shapes of the GIA fingerprints are significantly affected by the choice of the deglaciation chronology and of the rheological profile. An idea of this sensitivity can be obtained comparing directly the results obtained here and based on ICE-5G (VM2) with those of SM19, who have adopted a realization of ICE-6G (VM5a) [57]. In [32], it is possible to visualize the GIA fingerprints for old model ICE-3G (VM1) of Tushingham and Peltier [54], which differs from the most recent ICE-*X* models for a significantly larger meltwater release from Antarctica. For a discussion about the relationship between the fingerprints shapes and the GIA models employed, see in particular Spada and Galassi [73], who have emphasized how the choice of the model affects the GIA corrections at tide gauges and, consequently, the estimate of secular sea-level rise. Of course, GIA models that are independent from the ICE-*X* models exist and have a central role in the literature, like those progressively developed at the National Australian University by Kurt Lambeck and colleagues (see Nakada and Lambeck [74], Lambeck et al. [75] and subsequent contributions). Some GIA fingerprints for one of these models are shown in [32].

Recent reviews [31,76] have indicated that the evolution of the GIA models has been considerable during last decades, because of the increased availability of proxy data constraining the history of sea level in the last thousand years [31,34]. Such evolution has also motivated efforts aimed at extracting

geophysical information from ensembles (or more often mini-ensembles) of GIA models, as done by e.g., [32,77–82]. The evolution of GIA models shall certainly continue in the future, in order to account for more realistic (possibly three-dimensional) descriptions of the Earth's rheology, to include new details of the history of the ice sheets and their distribution, to relax some simplifying assumptions in the theory behind the SLE, to further fine-tune the rotation theory, and to add new elements or new branches to the interactions diagram of Figure 1. Thus, although their general properties associated to the principle of mass conservation shall not change, the shape of the GIA fingerprints is certainly not given once and for all.

**Author Contributions:** G.S. and D.M. have equally contributed to the development of the theory, to the numerical experiments, and to the writing of the manuscript.

**Funding:** G.S. is funded by a FFABR (Finanziamento delle Attività Base di Ricerca) grant of MIUR (Ministero dell'Istruzione, dell'Università e della Ricerca) and by a research grant of Dipartimento di Scienze Pure e Applicate (DiSPeA) of the University of Urbino "Carlo Bo".

**Acknowledgments:** Program SELEN[4] (SELEN version 4.0) is available from Zenodo at the link https://zenodo.org/record/3377404 (doi:10.5281/ zenodo.3339209) and from the Computational Infrastructure for Geodynamics (CIG) at *github.com/ geodynamics/selen*. The open source Love numbers calculator and Post Glacial Rebound Solver TABOO can be downloaded from https://github.com/danielemelini/TABOO. Some of the figures have been drawn using the Generic Mapping Tools (GMT) of Wessel and Smith [83]. We thank Gaia Galassi and Marco Olivieri for their advice and encouragement. We thank Francesco Mainardi for insightful discussion about the rheological aspects of GIA and for warm hospitality. G.S. has benefited from the serene atmosphere of the Naturalistic Annex of the Museum of Bagnacavallo (RA), Italy, where this paper has been conceived. Raffaello Mascetti has patiently revised the manuscript during various stages of its development, providing constructive comments and invaluable inspiration.

**Conflicts of Interest:** The authors declare no conflict of interest.

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
