# Peer review of "On Some Properties of the Glacial Isostatic Adjustment Fingerprints"

_water, doi:10.3390/w11091844_

Round 1
Reviewer 1 Report
On some properties of the Glacial Isostatic Adjustment fingerprints
Giorgio Spada and Daniele Melini
Summary
This study examines the fingerprints of the observables of GIA such as relative sea level rise, vertical displacement and geoid height variation calculated using a new version of the SELEN open source model. The authors relate the patterns of the fingerprints to physical processes and highlight for example the rotational and long wavelength components. The authors calculate global averages of the different fingerprints to compare rotational theory. The results are interesting particularly when related back to the underlying sea level equation. The paper is well written and organised.
Broad Comments
Whilst the calculations and modelling in this paper are robust, I feel that it is not really explained why it is important to examine the different GIA-related fingerprints and what the motivation for the study is. I am not sure what is new or different about this work in the context of the existing literature. I would therefore suggest an additional opening paragraph which explains a bit more the motivation and the importance of the work done in this paper. The introduction as it stands leaps straight into the background of the sea level equation which is a bit heavy going so an extra paragraph would also alleviate this problem. There is not much discussion of how the fingerprints would change given a different ice/earth model combination apart from being briefly mentioned in the individual results sections in the results. Since only results using ICE5G are show here it would be useful to summarise the expected differences in the fingerprints given different deglacial histories/earth models/contemporary change in the conclusions. Rotational theory- there is a brief mention of “revised” and “traditional” rotation theory on line 263 and results in the table reflect the use of these different theories. I may have missed it in the text but it is not obvious what the difference between these is, and it would be useful to include a brief statement along these lines rather than only a reference to a paper.Specific Comments
Line 10 – I’d prefer a different word to “sluggish” – delayed/slow or similar Line 63 – reference to Figure 5 from Clark et al. I think it would be useful to reproduce that figure since it is also referred to again in section 4 where the results are discussed. Figure 1 – define r in the caption. The rheology symbol is given but not defined in the caption. Line 413/414 – useful to point the reader to compare Figure 6 with Figure 5 here.
Reviewer 2 Report
Dear editor,
The manuscript "On some properties of the Glacial Isostatic Adjustment fingerprints" by Giorgio Spada and Daniele Melini is a review on GIA modelling and their resulting computations on some of the GIA fingerprint observables on the sea level. The theory is discussed well and followed by a methodology to construct the fingerprints. This discussion is valuable for your journal. Therefore, I would advise to publish this manuscript after some minor corrections, listed below.
Minor corrections:
R1: page 2 line 52-52. sentence ends with "sea surface and (.....)." I believe the () are misplaced.
R2: page 3 line 79 : "the term of fingerprint (function)...The elastic", Reads a bit strange. Some editing is needed.
R3: page 7 line 176, please define L and L0 in the text.
R4: page 10 line 253: for reproducibility, could the Loading and tidal love numbers be listed in a table.
R5: page 10 line 259: "for a given an approximation", for an approximation or given an approximation, not both.
R6: page 11 line 307: "with < S_dot >^0 = -0.14 mm/yr " Table 1 says -0.05 mm/yr. Are we discussing different values, because this is then not clear.
R7: page 12 line 316: Table 1 says -0.05 mm/yr?
R8: page 14 line 388: "<G_dot>0 approx <U_dot>o suggested by the results of column a) Table 1", which would mean -0.05 approx -0.22, which is an order difference. Could you give an argument for this 'large' approximation.
